# The Influence of Multidrug-Resistant Bacteria on Clinical Outcomes of Diabetic Foot Ulcers: A Systematic Review

**DOI:** 10.3390/jcm10091948

**Published:** 2021-05-01

**Authors:** Gianmarco Matta-Gutiérrez, Esther García-Morales, Yolanda García-Álvarez, Francisco Javier Álvaro-Afonso, Raúl Juan Molines-Barroso, José Luis Lázaro-Martínez

**Affiliations:** Diabetic Foot Unit, Clínica Universitaria de Podología, Facultad de Enfermería, Fisioterapia y Podología, Universidad Complutense de Madrid, Instituto de Investigación Sanitaria del Hospital Clínico San Carlos (IdISSC), 28040 Madrid, Spain; gmatta@ucm.es (G.M.-G.); ygarci01@ucm.es (Y.G.-Á.); alvaro@ucm.es (F.J.Á.-A.); rmolines@ucm.es (R.J.M.-B.); diabetes@ucm.es (J.L.L.-M.)

**Keywords:** diabetic foot, foot infection, multidrug-resistant organisms, amputation, systematic review

## Abstract

Multidrug-resistant organism infections have become important in recent years due to the increased prevalence of diabetic foot ulcers and their possible consequences. This study aimed to systematically review and evaluate ulcer duration, healing time, hospital stay, amputation, and mortality rates in patients with diabetic foot ulcers caused by infection with multidrug-resistant organisms. PubMed, the Cochrane Library, and Web of Science were searched in May 2020 to find observational studies in English about the clinical outcomes of multidrug-resistant organism infection in diabetic foot ulcers. Eight studies met the inclusion criteria, and these studies included 923 patients. The overall methodological quality of the study was moderate. Ulcer duration was described in six studies, and there was no practical association with multidrug-resistant organisms. Two out of three studies reported a longer healing time in multidrug-resistant organism infections than in non-multidrug-resistant organism infections. Clinical outcomes included the duration of hospitalisation, surgeries, amputations, and deaths. Lower limb amputation was the most reported clinical outcome in the included studies, and was more prevalent in the multidrug-resistant organism infections. We concluded that there was not enough evidence that multidrug-resistant organisms hindered the healing of diabetic foot ulcers. In contrast to the clinical outcomes, multidrug-resistant organisms affect both amputation rates and mortality rates.

## 1. Introduction

Diabetic foot infection (DFI) is thought to increase the risk of developing an amputation by 155 times [1], increasing the mortality rate at 5 years to 57% in these patients [2]. Diabetic foot osteomyelitis is the most common type of DFI, occurring in approximately 20% of moderate infections and in up to 60% of severe infections [3].

Diabetic foot ulcer (DFU) infections have been reported to have a longer healing time than uninfected ulcers [4]. Furthermore, unfavourable outcomes are associated with late diagnosis and, therefore, delayed effective treatment [5]. Thus, following the updated recommendations of the guidelines on DFI [6,7,8], the selection of empirical antibiotic treatment should be based on the likely etiologic agent and their antibiotic susceptibilities, as well as the severity of the infection.

The most commonly isolated microorganism in osteomyelitis remains *Staphylococcus aureus*, although infections are usually polymicrobial [9] and may include microorganisms from the genera Streptococci, Enterococci, Enterobacteriaceae, and Pseudomonas [10]. Additionally, methicillin-resistant *S. aureus* has shown a prevalence of between 15% and 30% [11,12], as well as being identified as a risk factor for re-hospitalisation [13].

Multidrug-resistant organism (MDRO) infections have been implicated in several studies on DFUs [14], and they were present in up to 53% of patients [15]. Dai et al. [16] recently performed a meta-analysis and identified risk factors for the development of MDRO in DFUs, including ischaemic aetiology, larger ulcer size, more severe ulcer classification, osteomyelitis, previous history of antibiotic therapy, and hospitalisation.

For clinical outcomes that are caused by MDRO in DFU, several studies have taken into account events such as the time of ulcer evolution [15,17,18,19,20,21], amputation [20,21,22,23], death [18,22,23], and length of hospitalisation [18,19,22]. Depending on the study, some of these variables have or have not shown an association with this kind of infection, and its impact on DFU is not clear.

As described, MDRO infections have a relevant prevalence in DFU, and interest in this condition has increased in recent years, leading to a large number of scientific publications. However, to the best of our knowledge, there has not been a study that has summarised the clinical outcomes of MDRO infection in DFU. The aim of this systematic review is to evaluate the ulcer duration, healing time, hospital stay, amputation, and mortality rates of DFU with MDRO infection.

## 2. Methodology

This systematic literature review was performed according to the Preferred Reporting Items for Systematic Review and Meta-Analysis (PRISMA) guidelines (Appendix A) [24].

### 2.1. Search Strategy

The literature search was performed using three electronic databases (PubMed, Cochrane Library, and Web of Science) in May 2020. The keywords used for the search were as follows: (‘multidrug resistant’ OR ‘multi-drug resistant’ OR ‘multidrug-resistance’) AND (diabetic foot). The search was limited to all articles that were published in English, beginning with the first article in July 2004 and concluding with the final article in May 2020, and studies that were conducted on humans.

### 2.2. Selection of Studies

The inclusion criteria were as follows: (1) observational study, case–control, and cohort studies in adult patients; (2) DFUs with multi-drug organism infection; (3) mean ulcer duration and the healing time of the ulcer were reported; and (4) evaluation of clinical outcomes related to infection. The exclusion criteria were as follows: (1) randomised controlled trials; (2) an alternative treatment included in addition to the standard care of the infected wound (antibiotic therapy and off-loading); (3) no data available for the analysis; and (4) non-original articles, including reviews, case reports, letters, or comments.

Randomised controlled trials were excluded to avoid any type of intervention that could disrupt the clinical outcome measures.

The articles that met any of our exclusion criteria were excluded by reading the abstract. Duplicate articles were also excluded. The full text of the rest of the articles was read to determine if they met the inclusion criteria. The review was performed independently by two authors (GMG and EGM). Any disagreements were resolved by discussion between both authors.

### 2.3. Data Collection

Data from the chosen studies were extracted by two authors (GMG and EGM) using a customised Excel spreadsheet. The data extracted from each article were as follows: main author, year, study design, country, age and number of participants, definition of MDRO, ulcer duration, time to heal, duration of hospitalisation, and prevalence of amputation and death related to the ulcer and infection (not on the patient’s history).

### 2.4. Quality Assessment

The evidence and recommendation levels were also evaluated according to the Oxford Centre for Evidence-based Medicine criteria (March 2009) [25].

The included studies were evaluated using a Strengthening the Reporting of Observational studies in Epidemiology (STROBE) checklist [26]. This guideline is a tool with 22 criteria to evaluate the study design quality and biases in the study. For each criterion that was met, the study is awarded one point; a higher score indicates a higher study quality.

## 3. Results

There were 217 articles that were identified in our systematic search. After the screening process, eight studies were included for the final analysis (Figure 1).

Table 1 summarises the design of the included studies, level of evidence, and degree of recommendation according to the Oxford criteria. Among the selected studies, six were retrospective cohorts [15,18,19,20,21,22] and two were prospective cohorts [17,23]. The percentage of adequate items from the STROBE checklist was 64.2% (Table 2).

These studies included 923 patients, ranging from a sample size of 65 [21] to 188 [23]. The MDRO infection rate was 17.8% [17] to 72.5% [18]. Ulcer duration was reported in six studies [15,17,18,19,20,21]; it exceeded 10 weeks in several studies [17,18,19,20] and only reached 25 weeks in one study [19]. Healing time was reported in only three studies [17,20,23], among which only one found significant differences [20].

Among the complications of DFU with MDRO infection, the duration of hospitalisation was reported in three studies [18,19,22], and a relationship was found in only two of these studies [19,22]. Amputation was the clinical outcome that was most often described [18,20,21,22,23], and there was an association with MDRO in all the studies where it was analysed. Death was analysed in three studies [18,22,23], but only one found an association with MDRO infection [22]. Table 3 summarises the characteristics of the studies that were included in this systematic review.

The definitions of MDRO that were used in the included studies were very different. The most frequently described definition was that bacteria were resistant to different antimicrobial agents or categories [15,17,19,20,22,23], followed by methicillin-resistant *S. aureus* [17,18,19,22,23] and extended spectrum beta-lactamase-producing bacteria [17,18,19,22]. The definitions of MDRO from each study are presented in Table 4.

## 4. Discussion

Generally, there was no evidence of an association between an increase in the ulcer duration and the presence of MDRO. However, two studies [20,23] indicated that the healing time was longer in MDRO infections than in non-MDRO infections. For clinical outcomes, this type of infection has an effect on lower limb amputations. 

Six [15,18,20,21,22,23] of the eight included studies were conducted in hospitals. When comparing infection rates, Gadepalli et al. [18] reported an MDRO infection prevalence of 72%, which is the highest among all the included articles. This study was conducted in a tertiary hospital in India. The studies by Noor et al. [21] and Zubair et al. [22] showed an MDRO infection rate of 60% and 46%, respectively, and both studies were also performed in India. However, several authors [18,20,21,22] associated the prevalence data with the widespread and prolonged use of broad-spectrum antibiotics, which leads to a selective survival benefit for pathogens. Zhang et al. [20] justified that infections caused by multidrug-resistant *Pseudomonas aeruginosa* are related to the prolonged antimicrobial treatment that is necessary for the antibiotic to penetrate into the *Pseudomonas* biofilm [27]. The two studies [17,23] with a lower prevalence of MDRO infection rate were performed on the European continent, which is in contrast to studies from other geographic areas with warm climates (e.g., India [18,21,22], China [15,20], and Turkey [19]), which showed a higher prevalence of resistant microorganisms such as Gram-negative bacteria, especially *P. aeruginosa* [28,29]. The study by Richard et al. [23], which is the penultimate study with the lowest prevalence, did not include patients with a previous history of antibiotic therapy in the last 6 months. This could explain some of their discrepant results compared to the rest of the studies. This antibiotic-free period has been standardised to 2 weeks to avoid false-negative cultures [8]. However, in many cases, this criterion can be difficult to meet because of the DFI severity.

Six studies [15,17,18,19,20,21] reported data on the mean ulcer duration, among which only one study [21] showed significant differences between the MDRO+ and the MDRO− groups. These results were unexpected because we thought that these studies would indicate a longer ulcer duration when a MDRO infection was present. The main differences between the study by Noor et al. [21] and the other studies is that it had the smallest sample size and was one of those with the highest prevalence of MDRO (56.92%). Another important aspect to consider is the retrospective nature of the studies, and most of the information on this variable was obtained from medical records. However, the study that reported an ulcer duration of 25 weeks [19] showed data with greater similarity to another study that only included ulcers with mild infection (mean, 22.4 weeks) [30].

From this review, the relevance of using DFU classifications has been investigated. The included articles used different classifications for ulcers, and the Wagner classification was used most frequently [15,17,18,20,21], although it does not take into account the presence of ischaemia. None of the included articles used the classifications to differentiate how ischaemia affected healing time. The presence or absence of ischaemia and the depth or degree of infection of the ulcer were related to a worse prognosis [8,31,32]. Thus, these variables require further study based on a more homogeneous population. 

Zhang et al. [20] was the only study that found significant differences in the healing time between the MDRO+ group and MDRO− group after 12 weeks of follow-up. The authors reported that it could be related to the exoU toxin, which was present in 70% of the *P. aeruginosa* in the MDRO+ group. This gene has been related to a higher proportion of multidrug resistance [33], and it also has a greater clinical impact on pneumonia because of its high virulence [34]. However, ulcers from the MDRO+ group, in the studies by Hartemann et al. [17] and Richard et al. [23], showed a higher prevalence of characteristics that were associated with a worse prognosis, such as osteomyelitis [3] and more severe infections, using the Infectious Diseases Society of America/International Working Group on the Diabetic Foot (IDSA/IWGDF) classification [8,35,36].

MDRO was associated with an increase in the duration of hospitalisation in two studies [19,22]. However, Gadepalli et al. [18] did not find statistically significant differences, and they also showed the shortest duration of hospitalisation. The authors state that this could be because of the political management and consequent aim of each hospital to discharge the patient after the ulcer had healed. Clarifying the association between this event and an infection by resistant bacteria is important because of the associated healthcare costs [37].

Amputation had a higher prevalence when there was a coexisting MDRO infection [18,20,21,22,23]. Despite this result, most of these studies only performed univariate statistical analyses. Zhang et al. [20] suggested that this result was related to the immunocompromised condition of patients with DFU, together with the ExoU toxin secreted by *P. aeruginosa*, which can quickly necrotize tissues and allow the development of osteomyelitis through the spread of *P. aeruginosa* into deep tissues. The rest of the authors [21,22,23] suggest that this increase in amputation rate may also be related to advanced diabetes, as well as the inadequate standard treatment of DFU. Richard et al. [23] stated that it was not possible for MDRO alone to explain the increased amputation rate in these patients.

Similar results were found for death, with only the study by Zubair et al. [22] reporting an association between death and the MDRO+ group. In this case, the multivariate analysis was also not performed, although there are studies that had related mortality to resistant microorganisms [38,39]. The only rationale that was mentioned for this clinical outcome was a relationship to poorly controlled blood glucose levels [22], concluding that the relationship between MDRO infection and mortality needs further investigation.

There were some limitations when interpreting the results in the current review. An important limitation is that most of the studies that have been performed to date on this topic are retrospective investigations that have focused on risk factors associated with the development of MDRO infections and not on clinical outcomes, making it difficult to obtain data for some variables. Another limitation of the included studies was the methodological and statistical heterogeneity, which led to results that had limited external validity. The STROBE checklist showed that the methodological quality of the studies can be improved. Moreover, having homogeneous data could provide more accurate information about the influence that MDRO has on patients with DFU.

There was wide heterogeneity regarding the definition of MDRO that was used in the included studies. Four authors [17,18,19,23] opted for very specific definitions, which were consistent with the two studies with the lowest prevalence. The study that was performed by Noor et al. [21] did not include a definition for this variable. The remaining investigations [15,20,22] from the last 10 years used a broader definition that had a greater similarity to that used by Ji et al. [15] This last study opted for a definition that was proposed by international experts who aimed to standardise the term MDRO, among others, in 2012 [40].

The heterogeneity of the study results that were included in the review does not allow a meta-analysis to be performed because it was not possible to compare the outcomes.

The main strength of our systematic review is that it is the first study that summarises the clinical outcomes that are derived from a MDRO infection in DFU. For the unanswered questions and future directions, prospective studies are needed to jointly analyse the effects of these microorganisms on the clinical outcomes in patients with DFU using a well-structured methodology and taking into account the possible biases that have been described above.

## 5. Conclusions

Currently, there is not enough scientific evidence to conclude that MDRO affects the healing time of ulcers. However, amputations and mortality rates have been affected by this type of infection.

## Figures and Tables

**Figure 1 jcm-10-01948-f001:**
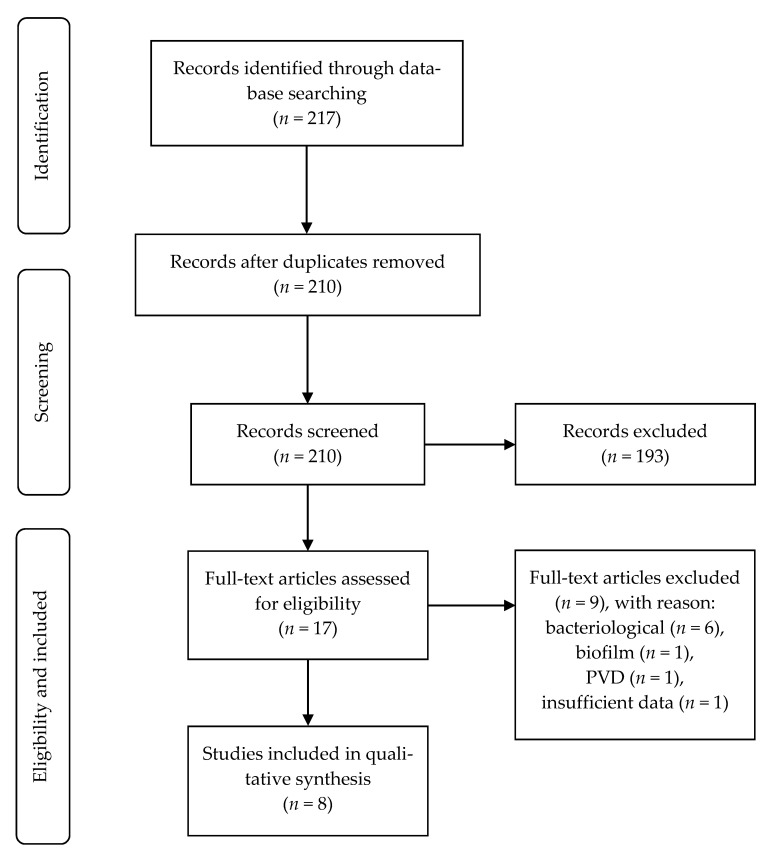
STROBE flow chart. Abbreviation: PVD, peripheral vascular disease.

**Table 1 jcm-10-01948-t001:** Levels of evidence and degree of recommendation for the included studies.

	Study Design	Level of Evidence	Degree of Recommendation
Ji et al. [15]	Retrospective cohort	2b	B
Hartemann et al. [17]	Prospective cohort	1b	A
Gadepalli et al. [18]	Retrospective cohort	2b	B
Kandemir et al. [19]	Retrospective cohort	2b	B
Zhang et al. [20]	Retrospective cohort	2b	B
Noor et al. [21]	Retrospective cohort	2b	B
Zubair et al. [22]	Retrospective cohort	2b	B
Richard et al. [23]	Prospective cohort	1b	A

**Table 2 jcm-10-01948-t002:** Rating for the Strengthening the Reporting of Observational studies in Epidemiology (STROBE) checklist.

	Item Number–STROBE Guidelines																						
	1		2	3	4	5	6	7	8	9	10	11	12	13	14	15	16	17	18	19	20	21	22
	T	A																					
Ji et al. [15]	N	Y	Y	N	N	Y	Y	Y	Y	N	N	N	N	Y	Y	Y	Y	N	Y	Y	Y	N	Y
Hartemann et al. [17]	N	Y	Y	N	N	Y	Y	Y	Y	Y	N	Y	N	Y	Y	Y	N	Y	Y	N	Y	Y	N
Gadepalli et al. [18]	N	Y	N	N	N	Y	N	Y	Y	N	N	Y	N	N	Y	Y	Y	N	N	N	Y	N	Y
Kandemir et al. [19]	N	Y	Y	N	N	Y	Y	Y	Y	Y	N	Y	Y	Y	Y	Y	N	Y	Y	N	Y	Y	N
Zhang et al. [20]	N	Y	Y	N	Y	Y	Y	Y	Y	N	N	Y	N	Y	Y	Y	Y	Y	Y	Y	Y	Y	Y
Noor et al. [21]	N	Y	Y	N	Y	Y	Y	Y	Y	N	N	Y	N	Y	Y	Y	N	N	Y	N	Y	Y	Y
Zubair et al. [22]	N	Y	Y	N	Y	Y	Y	Y	Y	N	N	Y	N	Y	Y	Y	Y	Y	N	N	Y	Y	Y
Richard et al. [23]	N	Y	N	N	Y	Y	Y	Y	Y	N	N	Y	Y	Y	Y	Y	Y	Y	Y	N	Y	Y	Y

Abbreviations: T, Title; A, Abstract; N, No; Y, Yes.

**Table 3 jcm-10-01948-t003:** Characteristics of the studies that were included in the systematic review.

Study/Year	Country	Participants Age N = MDRO+/MDRO−	Ulcer Duration MDRO+/MDRO− *p*-Value	Time to Ulcer HealingMDRO+/MDRO− *p*-Value	Event MDRO+/MDRO− *p*-Value	
Ji et al. [15]/2014	China	64.1 ± 10.5/64.4 ± 11 118 = 64/54	8.54 ± 8.43/5.96 ± 5.94 0.053	No description	No description	
Hartemann et al. [17]/2004	France	65 ± 12 180 = 32/148	19.71 ± 36/28.71 ± 55.71 0.30	Survival analysis No include mean ± SD 0.71	No description	
Gadepalli et al. [18]/2006	India	53.9 ± 12.1 80 = 58/22	12.43 ± 9.04/15.86 ± 11.14 0.14	No description	Duration of hospitalisation	2.76 ± 0.88/2.65 ± 0.8 0.61
					Surgery (amputation included)	47 (81)/10 (45.5) **<0.01** *
					Death	2 (3.4)/0 (0.0) *0.38*
Kandemir et al. [19]/2006	Turkey	60 ± 11 73 = 36/37	25.71 ± 77.43/16.14 ± 28.43 0.32	No description	Duration of hospitalisation	5.29 ± 4.14/2.86 ± 2.71 **0.00** **
Zhang et al. [20]/2014	China	65 ± 12.3/64 ± 10.8 117 = 43/74	10.6 ± 8.8/9.4 ± 6.3 0.457	Follow up 12 weeks 9 (20.9)/31 (41.9) **0.032**	Amputation	14 (32.6)/12 (16.2) **0.032**
Noor et al. [21]/2017	India	53 ± 9 65 = 37/28	9 ± 6.86/6.86 ± 3.43 <0.05 **	No description	Amputation	19 (51.35)/7 (25) <**0.05** **
Zubair et al. [22]/2010	India	49.11 ± 12.46 102 = 46/56	No description	No description	Duration of hospitalisation	3.84 ± 1.03/2.56 ± 0.41 **0.005** **
					Amputation	19 (41.3)/4 (7.1) **<0.001** **
					Death	4 (8.6)/1 (1.7) ***0.002*** **
Richard et al. [23]/2008	France	68.0 (no include SD) 188 = 45/143	No description	Follow up 10 weeks 25 (51.1)/99 (69.2) **0.04** * Survival analysis 14/10 (Does not include SD) **0.036** *	Total amputation	16 (35.6)/16 (11.2) **<0.001** **
					Major amputation	3 (18.8)/1 (6.3) **0.05** **
					Minor amputation	13 (81.2)/15 (93.7) **0.02** **
					Death	2 (4.4)/8 (5.6) NS

Data are presented as the mean ± SD (weeks) (%). Abbreviations: MDRO: multidrug-resistant organism; MDRO+: positive multidrug-resistant organism group; MDRO−: negative multidrug-resistant organism group; NS, not significant; SD, standard deviation; *, only significant in the univariate analysis; **, authors did not perform a multivariate analysis. The statistical analysis concerns original data from the articles included in the systematic review.

**Table 4 jcm-10-01948-t004:** Definition of multidrug-resistant organisms that was chosen by each study.

	Definition of MDRO
Ji et al. [15]	Bacteria resistant to at least one agent in three or more antimicrobial categories.
Hartemann et al. [17]	MRSA; bacteria producing extended spectrum ESBL; *Pseudomonas aeruginosa* resistant to both ceftazidime and imipenem; *Acinetobacter baumannii* resistant to imipenem.
Gadepalli et al. [18]	MRSA; ESBL producing bacteria; methicillin-resistant coagulase-negative *Staphylococci*.
Kandemir et al. [19]	MRSA; methicillin-resistant *Staphylococcus epidermidis*; penicillin resistant *Staphylococcus pneumoniae*; *Enterococcus* spp.; ESBL producing bacteria and inducible beta-lactamase; *P. aeruginosa* resistant to both ceftazidime and imipenem; *A. baumannii* resistant to imipenem.
Zhang et al. [20]	Bacteria resistant to at least 1 agent in each of the 3 or more antipseudomonal agents.
Noor et al. [21]	No description.
Zubair et al. [22]	Bacteria resistant to two or more antimicrobial classes; MRSA; ESBL producing organisms.
Richard et al. [23]	MRSA, *Enterobacteriaceae* resistant to third-generation cephalosporins; *P. aeruginosa* resistant to two antibiotics from among ticarcillin, ciprofloxacin, ceftazidime and imipenem; *Enterococcus* spp. resistant to glycopeptides; *A. baumannii* resistant to ticarcillin; *Stenotrophomonas maltophilia*.

Abbreviations: MRSA, methicillin-resistant *Staphylococcus aureus*; ESBL, extended spectrum beta-lactamase.

## Data Availability

The data are available previous request to corresponding author.

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
