# Peer review of "The Influence of Multidrug-Resistant Bacteria on Clinical Outcomes of Diabetic Foot Ulcers: A Systematic Review"

_jcm, 2021, doi:10.3390/jcm10091948_

Round 1
Reviewer 1 Report
Please see my notes attached to the PDF file.

Author Response
Thank you very much for your review. Following we are answering concerns that you have detailed in your review.
I would recommend the authors to add a few review articles to address the antibiotics used clinically to treat DFU infections. For example: DOI: 10.1021/acs.accounts.0c00864. Authors should include other reviews as well.
Thank you, we add the next paragraph following your recommendation, lines 34-39:
Diabetic foot ulcer (DFU) infections have been reported to have a longer time to healing than uninfected ulcers [4]. Furthermore, unfavorable outcomes are associated with late diagnosis and, therefore, delayed effective treatment [5]. Thus, following the updated recommendations of the guidelines on DFI [6–8], the selection of empirical antibiotic treatment should be based on the likely etiologic agent and their antibiotic susceptibilities, as well as the severity of the infection.
- Chang M, Nguyen TT. Strategy for Treatment of Infected Diabetic Foot Ulcers. Acc Chem Res 2021;54:1080–93. https://doi.org/10.1021/acs.accounts.0c00864.
- van Baal JG. Surgical Treatment of the Infected Diabetic Foot. Clin Infect Dis 2004;39:S123–8. https://doi.org/10.1086/383273.
- Lipsky BA, Aragón-Sánchez J, Diggle M, Embil J, Kono S, Lavery L, et al. IWGDF guidance on the diagnosis and manage-ment of foot infections in persons with diabetes. Diabetes Metab Res Rev 2016;32:45–74. https://doi.org/10.1002/dmrr.2699.
- Lipsky BA, Senneville É, Abbas ZG, Aragón‐Sánchez J, Diggle M, Embil JM, et al. Guidelines on the diagnosis and treatment of foot infection in persons with diabetes (IWGDF 2019 update). Diabetes Metab Res Rev 2020;36:e3280. https://doi.org/10.1002/dmrr.3280.
- Lipsky BA, Berendt AR, Cornia PB, Pile JC, Peters EJG, Armstrong DG, et al. 2012 Infectious Diseases Society of America Clinical Practice Guideline for the Diagnosis and Treatment of Diabetic Foot Infections. Clin Infect Dis 2012;54:e132–73. https://doi.org/10.1093/cid/cis346.
What it the starting time point from XXXX to May 2020?
Thank you very much for your comment. According to your recommendation we added the following sentence in lines 75-77: “The search was limited to all articles that were published in English from the first article, July 2004, that was published until May 2020 and studies that were conducted on humans”
Please define ulcer evolution?
Please be specific? What does the author mean by ulcer evolution time?
As it gets worse and bigger in wound size?
Thank you very much for your advice. To clarify this issue we have change ulcer evolution for ulcer duration. Line 100 and 149.
How many are few?
Thank you very much for your advice. According to your suggestion we re-write the following sentence, in lines 169-171: “However, two studies [20,23] have been found to support that the time to healing is longer in MDRO infections than in non-MDRO infections. For clinical outcomes, this type of infection has an effect on lower limb amputations.”.
- Zhang J, Chu Y, Wang P, Ji X, Li X, Wang C, et al. Clinical Outcomes of Multidrug Resistant Pseudomonas aeruginosa Infection 251 and the Relationship With Type III Secretion System in Patients With Diabetic Foot. Int J Low Extrem Wounds 2014;13:205–10. 252 https://doi.org/10.1177/1534734614545878.
- Richard J-L, Sotto A, Jourdan N, Combescure C, Vannereau D, Rodier M, et al. Risk factors and healing impact of multidrug-resistant bacteria in diabetic foot ulcers. Diabetes Metab 2008;34:363–9. https://doi.org/10.1016/j.diabet.2008.02.005.
Please expand on this, what authors mean by DFU healing? healing time or healing grade? Because the wounds with infection would get worse (higher in Wagner Grade) before they become amputated
Thank you very much for your comment. Taking account your advice, we have change the conclusion in lines 267-269: “Currently, there is not enough scientific evidence to conclude that MDRO affects to time to ulcer healing. However, the prognosis related to amputations and mortality has been affected by this type of infection”
Reviewer 2 Report
This is a well-done systematic review worthy of publication. I have some suggestions to improve the manuscript.
- Lines 42-43: “Multidrug-resistant organism (MDRO) infections have shown concerning data regarding the diabetic foot ulcers (DFU) in several studies…” This needs rewriting to make sense, something like: Multidrug-resistant organism (MDRO) infections have been implicated in several studies regarding diabetic foot ulcers (DFUs)…
- Line 46: “higher of ulcer classification” Suggest something like “more severe ulcer classification”
- Inclusion and exclusion criteria: please provide a rationale as to why control groups of randomized controlled trials were excluded (I assume these are excluded; they are not observational studies).
- Line 86: time of ulcer evolution, time to healing, time of hospitalization. What is time of ulcer evolution? Is it ulcer duration (how long it has existed)? Would it be better to describe time of hospitalization as duration of hospitalization, unless the authors meant how long since ulcer formation it took to be hospitalized? Please standardize any new definitions through the manuscript.
- Figure 1: insufficient, not unsufficient
- Line 110: mean ulcer healing. What does this mean? Proportion of wounds healed (at time X?); percentage area reduction (PAR)? Something else? Why would this be of interest, since it is not on the list for data extraction?
- Table 3: “ceftazidima e imipenem” This needs translation into English. There are several other instances where Spanish phrases need translation into English. Please fix.
- Line 135: 17.78%. one decimal place is sufficient, i.e., 17.8%
- Table 4: Please ensure the key to abbreviations include all abbreviations—i.e., BMR and in this context what + and – mean.
- I assume that the studies were too dissimilar in design and in outcome data to consider meta-analysis. If so, please add a sentence toward the end of the Discussion to describe this point.
- Lines 199-200: “Moreover, having homogeneous data could provide more accurate information about the influence that MDRO has on patients with DFU.” Do the authors have any suggestions for the best type of outcomes, study lengths, etc.?
- I might suggest that the authors make available the filled-in PRISMA checklist.
Author Response
Thank you very much for your review. Following we are answering concerns that you have detailed in your review.
Reviewer 2:
- Lines 42-43: “Multidrug-resistant organism (MDRO) infections have shown concerning data regarding the diabetic foot ulcers (DFU) in several studies…” This needs rewriting to make sense, something like: Multidrug-resistant organism (MDRO) infections have been implicated in several studies regarding diabetic foot ulcers (DFUs)…
Thank you, following your advice, we re-write the next sentence, in lines 48-50:
“Multidrug-resistant organism (MDRO) infections have been implicated in several studies regarding diabetic foot ulcers (DFU) [14], and they were present in up to 53% of patients [15].”
- Chen Y, Ding H, Wu H, Chen H-L. The Relationship Between Osteomyelitis Complication and Drug-Resistant Infection Risk in Diabetic Foot Ulcer: A Meta-analysis. Int J Low Extrem Wounds 2017;16:183–90. https://doi.org/10.1177/1534734617728642.
- Ji X, Jin P, Chu Y, Feng S, Wang P. Clinical Characteristics and Risk Factors of Diabetic Foot Ulcer With Multidrug-Resistant Organism Infection. Int J Low Extrem Wounds 2014;13:64–71. https://doi.org/10.1177/1534734614521236.
- Line 46: “higher of ulcer classification” Suggest something like “more severe ulcer classification”
Thank you, following your advice, we have changed the words used, in lines 50-53:
Dai et al.[16] recently performed a meta-analysis and identified risk factors for the development of MDRO in DFU including ischaemic aetiology, larger ulcer size, more severe ulcer classification, osteomyelitis, previous history of antibiotic therapy, and hospitalisation.
- Dai J, Jiang C, Chen H, Chai Y. Assessment of the Risk Factors of Multidrug-Resistant Organism Infection in Adults With Type 1 or Type 2 Diabetes and Diabetic Foot Ulcer. Can J Diabetes 2019:1–8. https://doi.org/10.1016/j.jcjd.2019.10.009.
- Inclusion and exclusion criteria: please provide a rationale as to why control groups of randomized controlled trials were excluded (I assume these are excluded; they are not observational studies).
Thank you, we add the next lines following your recommendation in lines 86-88: “The control groups of the randomized controlled trials were excluded in order to avoid any type of intervention that could disrupt the outcome measures”
- Line 86: time of ulcer evolution, time to healing, time of hospitalization. What is time of ulcer evolution? Is it ulcer duration (how long it has existed)? Would it be better to describe time of hospitalization as duration of hospitalization, unless the authors meant how long since ulcer formation it took to be hospitalized? Please standardize any new definitions through the manuscript.
Thank you very much, we agree with you, so we have corrected the words through the manuscript and used “ulcer duration” and “duration of hospitalisation” instead of “time of ulcer evolution” and “time of hospitalisation”.
- Figure 1: insufficient, not unsufficient
Thank you, following your advice, we have changed the word for “insufficient” used on figure 1.
- Line 110: mean ulcer healing. What does this mean? Proportion of wounds healed (at time X?); percentage area reduction (PAR)? Something else? Why would this be of interest, since it is not on the list for data extraction?
Thank you very much for your comment. We have revised the sentence and is due to a typo error. We have changed the term "mean ulcer healing" for "ulcer duration”. Line 149
- Table 3: “ceftazidima e imipenem” This needs translation into English. There are several other instances where Spanish phrases need translation into English. Please fix.
Thank you very much for your comment. We have reviewed the table 3:
We have corrected the translation mistakes,
- “ceftazidima e imipenem” to “ceftazidime and imipenem”;
- “bacteria-producing ESBL” to “ESBL producing bacteria”;
- “bacteria producing ESBL and inducible beta-lactamase” to “ESBL producing bacteria and inducible beta-lactamase”;
- “Bacteria resistant to at least 1 agent in each of the 3 or more classes of antipseudomonal agents” to “Bacteria resistant to at least 1 agent in each of the 3 or more antipseudomonal agents”;
- “bacteria resistant to two or more classes of antimicrobials” to “bacteria resistant to two or more antimicrobial classes”.
- Line 135: 17.78%. one decimal place is sufficient, i.e., 17.8%
Thank you for your recommendation, we have corrected to 17.8%. Line 182
- Table 4: Please ensure the key to abbreviations include all abbreviations—i.e., BMR and in this context what + and – mean.
Thank you, we add the key to new abbreviations on the table 4:
MDRO: multidrug-resistant organism; MDRO+: positive multidrug-resistant organism group; MDRO-: negative multidrug-resistant organism group. Table 4.(Lines 195-196)
- I assume that the studies were too dissimilar in design and in outcome data to consider meta-analysis. If so, please add a sentence toward the end of the Discussion to describe this point.
Thank you, we add the next paragraph following your recommendation, lines 255-257.
The heterogeneity of the studies included in the review regarding the results does not allow a meta-analysis to be carried out, since there was no possibility of comparing the outcomes.
- Lines 199-200: “Moreover, having homogeneous data could provide more accurate information about the influence that MDRO has on patients with DFU.” Do the authors have any suggestions for the best type of outcomes, study lengths, etc.?
Thank you, we add the next paragraph following your recommendation, lines 260-264.
“Regarding unanswered questions and future directions, we consider the need for prospective studies with the aim of analysing the effects of this type of microorganisms on the healing and prognosis of patients with DFU has been clarified, on the basis of a well-structured methodology and taking into account the possible biases that have been described above.”
- I might suggest that the authors make available the filled-in PRISMA checklist.
Thank you for your suggestion, we agree with you and we have prepared the checklist file annex.
